# The *Dct^−/−^* Mouse Model to Unravel Retinogenesis Misregulation in Patients with Albinism

**DOI:** 10.3390/genes13071164

**Published:** 2022-06-27

**Authors:** Angèle Tingaud-Sequeira, Elina Mercier, Vincent Michaud, Benoît Pinson, Ivet Gazova, Etienne Gontier, Fanny Decoeur, Lisa McKie, Ian J. Jackson, Benoît Arveiler, Sophie Javerzat

**Affiliations:** 1Rare Diseases Genetics and Metabolism, INSERM U1211, SBM Department, University of Bordeaux, F-33076 Bordeaux, France; angela.tingaud-sequeira@u-bordeaux.fr (A.T.-S.); elina.mercier@etu.u-bordeaux.fr (E.M.); vincent.michaud@chu-bordeaux.fr (V.M.); benoit.arveiler@chu-bordeaux.fr (B.A.); 2Molecular Genetics Laboratory, Bordeaux University Hospital, F-33076 Bordeaux, France; 3SAM, TBMcore, CNRS UAR 3427, INSERM US005, Université Bordeaux, F-33076 Bordeaux, France; pinson@ibgc.cnrs.fr; 4MRC Human Genetics Unit, University of Edinburgh, Edinburgh EH4 2XU, UK; ivet.gazova@gmail.com (I.G.); lisa.mckie@ed.ac.uk (L.M.); ian.jackson@ed.ac.uk (I.J.J.); 5Bordeaux Imaging Center, CNRS, INSERM, BIC, UMS 3420, US 4, University Bordeaux, F-33076 Bordeaux, France; etienne.gontier@u-bordeaux.fr (E.G.); fanny.decoeur@u-bordeaux.fr (F.D.)

**Keywords:** albinism, retinal pigment epithelium, DCT, melanogenesis, melanosomes, L-Dopa

## Abstract

We have recently identified *DCT* encoding dopachrome tautomerase (DCT) as the eighth gene for oculocutaneous albinism (OCA). Patients with loss of function of *DCT* suffer from eye hypopigmentation and retinal dystrophy. Here we investigate the eye phenotype in *Dct^−/−^* mice. We show that their retinal pigmented epithelium (RPE) is severely hypopigmented from early stages, contrasting with the darker melanocytic tissues. Multimodal imaging reveals specific RPE cellular defects. Melanosomes are fewer with correct subcellular localization but disrupted melanization. RPE cell size is globally increased and heterogeneous. P-cadherin labeling of *Dct^−/−^* newborn RPE reveals a defect in adherens junctions similar to what has been described in tyrosinase-deficient *Tyr^c/c^* embryos. The first intermediate of melanin biosynthesis, dihydroxyphenylalanine (L-Dopa), which is thought to control retinogenesis, is detected in substantial yet significantly reduced amounts in *Dct^−/−^* postnatal mouse eyecups. L-Dopa synthesis in the RPE alone remains to be evaluated during the critical period of retinogenesis. The *Dct^−/−^* mouse should prove useful in understanding the molecular regulation of retinal development and aging of the hypopigmented eye. This may guide therapeutic strategies to prevent vision deficits in patients with albinism.

## 1. Introduction

Albinism is a phenotypically and genetically heterogeneous disease affecting ~1/17,000 individuals worldwide [1]. It is commonly thought of as solely a pigmentation anomaly. However, while skin and hair hypopigmentation is highly variable, from extreme to undetectable, patients with confirmed albinism systematically suffer from very poor vision, albinism being the second most frequent cause of congenital blindness [2]. Early developmental anomalies such as excessive optic nerve decussation and foveal hypoplasia are the major causes of low vision in these patients for whom severe iris/fundus hypopigmentation is detected even in the absence of a skin and/or hair phenotype [3,4]. Several pieces of experimental evidence suggest that the embryonic retinal pigment epithelium (RPE) is central to the phenotype in that defective melanogenesis in this cell layer initiates a cascade of detrimental events responsible for the disrupted development of the adjacent sensory retina. The first clues date back to our own observations in the early 1990 s when, by using genetic ablation of the embryonic RPE at the onset of pigmentation, we could show that the RPE layer is essential for retinal neurogenesis [5]. The importance of melanogenesis within the differentiating retina was elegantly evidenced further with a series of observations by Carol Mason’s group. Scrutinizing the standard *albino* mouse (*Tyr^c/c^*) with loss of function of the primary melanogenic enzyme tyrosinase, they showed that defects in melanogenesis within the embryonic eye alter the shape of the RPE cells and their communication with the adjacent neural retina at critical stages of photoreceptor and retinal ganglion cell fate [6,7,8,9]. In parallel, it soon appeared that, rather than the final pigment, one or several melanin intermediates in the RPE are required for orchestrating major steps of retinogenesis. By rescuing dihydroxyphenylalanine (L-Dopa) synthesis in the RPE of *albino* mouse transgenic embryos overexpressing tyrosine hydroxylase, Luis Montoliu’s lab was able to rescue normal photoreceptor counts as well as correct routing of the optic nerve fibers [10]. In wild-type embryos and newborns, the kinetics of L-Dopa production is compatible with such a paracrine role [9,11]. Thus, L-Dopa appears to be a strong candidate for inducing some if not all of the retinal processes that are deficient in albino patients.

We recently described *DCT/TYRP2* as a new gene involved in oculocutaneous albinism type 8 (OCA8) [12], raising the total number of albinism genes to 20 [13]. The two patients included in our study have a moderate cutaneous phenotype but typical hypoplastic fovea, iris transillumination, and fundus hypopigmentation. Soon after our initial description, more OCA patients were detected with pathogenic variants in *DCT* [14]. *DCT* encodes dopachrome tautomerase (DCT/TYRP2), a melanosomal enzyme that acts downstream of L-Dopa, converting dopachrome into 5,6-dihydroxyindole-2-carboxylic acid (DHICA) on the dark melanin pathway [15] (Figure 1). Intriguingly, patients with OCA8 and OCA3 (*TYRP1* variants) share ophthalmological deficits described in OCA1 (*TYR* variants), suggesting unknown epistatic interference.

Massive amounts of biochemical and physiological data have been collected from *TYR* and *TYRP1* loss-of-function models, both in cultured cells and in animal models, for review [13,21,22]. By contrast, the impact of the loss of function of *DCT* on pigmentation and even more so on ocular development has not been explored in such detail. For decades, only three spontaneous mouse mutant lines (*Slaty, Slaty-J,* and *Slaty Light)*, with amino acid substitution rather than null, were available [16]. In 2004, Guyonneau and colleagues designed a genetically engineered knockout model for *DCT* [23]. After analyzing adult mice by conventional histology, as well as derived melanocytes in culture, they concluded that only the skin and fur color were significantly affected, not the pigmentation and/or morphology of the eye tissues, i.e., iris, choroid, and RPE.

For our recent validation of OCA8, we generated two lines of *DCT* mutant mice harboring alleles equivalent to those identified in our patients, as well as a null control line, *Dct^−/−^* [12]. Focusing our attention on the RPE, we could show that the loss of function of *Dct* has major effects on the pigmentation of this critical cell layer in the adult. By contrast, the neighboring choroid and iris seemed barely affected, which may explain why the RPE defect had not been noticed in the original knockout line. Given the clear OCA profile of patients with pathogenic variants in *DCT* on the one hand and the RPE hypopigmentation of the mouse lines on the other hand, it is worthwhile to explore the *Dct^−/−^* mouse in more detail, especially in the perinatal period, and search for common and distinct phenotypes compared to other models of albinism such as *Tyr^c/c^* mice. These investigations will help in understanding the molecular link between melanogenesis in embryonic RPE and retinal development. They should have important implications for the prognosis of retinal pathology in patients according to their albinism type.

Here we carefully examine melanogenesis in *Dct^−/−^* mice with a specific focus on the RPE. Using multimodal cell imaging, we identify RPE defects, such as incomplete melanosome maturation, cell shape, and junctional anomalies, highlighting some features similar to what has been described for tyrosinase loss of function. We also show that L-Dopa is detected in substantial amounts in *Dct^−/−^* postnatal mouse eyecups, although significantly less than in controls, raising the question of its concentration in the embryonic RPE during retinogenesis.

## 2. Materials and Methods

### 2.1. Mice

*Dct^−/−^* mice were maintained at the University of Bordeaux (France). Breeding conditions and experimental procedures were approved by the local ethical committee (CE050). Genotyping of the *DCT* colony (raised on the *non-agouti* C57BL/6J background) was performed by PCR and sequencing on tail tip DNA using the Thermo Scientific^TM^ Phire^TM^ Tissue Direct PCR Master Mix with specific primers to detect the CRISPR-Cas9 engineered deletion (5 bp at nucleotide position 177-181) that results in loss of function as described [12]. Heterozygous females (*Dct^+/−^*) were crossed to homozygous mutant males (*Dct^−/−^*) to generate litters containing half mutants (*Dct^−/−^*) and half heterozygous mice (*Dct^+/−^*) of the same age. *Dct^+/−^* was chosen as an age-matched wild-type reference after the full recessivity of each studied phenotype had been carefully checked by comparing *Dct^+/−^* and coisogenic C57BL/6J *Dct^+/+^* samples (with the exception of L-Dopa dosage). Coisogenic *Tyr^c/c^* controls were obtained from Dr. Alexandra Rebsam (Institut de la Vision, Paris, France) [7].

### 2.2. Skin Flat mounts Analysis

Heads of P0.5 and P2 mice were fixed in 4% paraformaldehyde (PFA) for 1 h at room temperature. Head skins were rinsed and dissected in PBS, then soaked in increasing concentrations of glycerol and mounted in 100% glycerol, the inner side facing the coverslip. The images were taken with a Nikon Eclipse E1000 microscope. The number and pigmentation of hair follicles were estimated using the “Analyze Particles” command in ImageJ (https://imagej.nih.gov/ij/, accessed on 21 June 2022, RRID: nif-0000-30467). Statistical significance was tested by an unpaired *t*-test.

### 2.3. RPE Flat mounts Staining

Newborn heads were collected and fixed by immersion and gentle shaking in 4% PFA for 1 h at room temperature, then rinsed twice in PBS. The eyeball was perforated at the nasal pole of the retina for future orientation. The eye was removed, refixed in 4% PFA for 30 min, then transferred to PBS without calcium and magnesium. The cornea was cut under a dissecting microscope and the lens removed, as well as extraocular tissue. Eyecups were bleached by incubation for 30 min in 3% H_2_O_2_/0.5% KOH, then rinsed 3 times. Eyecups were then permeabilized and blocked in PBS with 1% Triton-X 100 and 1% BSA for one hour. For staining F-actin (filamentous actin), eyecups were incubated overnight at 4 °C with Alexa Fluor 488 Phalloidin (1:500; Invitrogen, Carlsbad, CA, USA) in PBS. The neural retina was carefully detached and discarded. Four radial cuts were made, and the RPE-choroid-sclera-tissue was mounted flat on a microscope slide using Fluoroshield^TM^ with DAPI (Sigma-Aldrich, St-Louis, MO, USA), the RPE facing the coverslip. Staining was visualized and imaged with a Nikon Eclipse E1000 microscope, using the 10× objective for topography and 40× for RPE cell morphology assessment. For P-cadherin detection, fixed and permeabilized eyecups were incubated with anti-P-cadherin rat monoclonal antibody (13-2000Z, Invitrogen, Carlsbad, CA, USA) in PBS containing 0.2% Triton-X 100 and 1% BSA at 1:500 dilution, overnight. After PBS washes, they were incubated for 1 h at room temperature with 488 donkey anti-rat secondary antibodies (Invitrogen, Carlsbad, CA, USA) at a 1:500 dilution. They were rinsed with PBS and mounted as described above. Staining was visualized and imaged with a Zeiss Axio-Observer Z1 microscope, using the 63× objective.

### 2.4. Transmission Electron Microscopy

Skin from P4 mouse heads was dissected and fixed in a cacodylate sodium buffer 0.1 M, pH 7.4 containing 2% PFA and 2.5% glutaraldehyde for 1 h at room temperature. For retinas, 1-month-old mice were culled, and the eyes were enucleated, punctured, and immediately fixed as above. Patches of tissues (2 mm^2^) were cut, washed in 0.1 M cacodylate buffer, then post-fixed with 1% osmium tetroxide for 2 h on ice in the dark. Samples were washed in pure water (RiOs-DI type II water purification system (resistivity > 10 MΩ cm)) and dehydrated in increasing concentrations of ethanol. They were infiltrated with a mixture of pure ethanol and epoxy resin 50/50 (*v/v*) (Epon 812; Delta Microscopies, Mauressac, France) for 2 h at RT and then embedded in pure resin overnight at RT. The polymerization of the resin was carried out at 60 °C for 48 h. Ultra-thin sections were cut at 65 nm using a Leica EM UC7 Ultramicrotome and placed on copper square mesh grids (cat#G150-Cu, Delta Microscopies, Mauressac, France), then stained with Uranyless (Delta Microscopies, Mauressac, France) and lead citrate. Pictures were acquired at a magnification ranging between 5 K and 60 K under a transmission electron microscope (H7650, Hitachi, Tokyo, Japan) at 80 Kv.

### 2.5. Melanosome Quantification and Staging

Melanosomes were identified and characterized (staging and shape) according to previously described criteria [24,25,26,27]. For quantification, melanosomes were scored on at least 4 independent areas of the tissue representing around 2 mm^2^ of cytoplasmic surface for each genotype. The cytoplasmic areas were delimited and measured using ImageJ on electron micrographs at 10 K magnification.

### 2.6. L-Dopa Dosage

Eyes were collected from euthanized mouse littermates at midmorning for each time point, soaked in PBS without calcium and magnesium at room temperature for 30 min, and dissected. Extraocular tissues were removed under a dissecting microscope, as well as the cornea and lens. The neural retina was gently detached and discarded. The two eyecups of each individual were soaked in 40 µL of 0.2 M perchloric acid and stored at −20 °C. Amine extraction was performed with 3 cycles of 30 s of sonication, a 21,000 g centrifugation for 10 min at 4 °C, followed by a second sonication/centrifugation step on the first pellet resuspended in 40 µL of perchloric acid. Supernatants were pooled and centrifuged at 21,000 g for 1 h at 4 °C. Extracted metabolites in the supernatant (50 µL) were separated on an ICS3000 chromatography station (Thermo Electron SAS, Waltham, MA, USA) using an IonPac CS16-Fast-4µ column (150 × 2 mm; Thermo Electron SAS). Separation was achieved in isocratic conditions with perchloric acid 10 mM for 60 min at 0.16 mL/min flow. L-Dopa, detected by UV absorbance at 280 nm using a diode array detector (Ultimate 3000 RS; Thermo Electron SAS, Waltham, MA, USA), was identified by its retention time, UV spectrum signature, and co-injection with standard and was quantified by comparison with pure compound.

## 3. Results

### 3.1. Dct^−/−^ Pups Are Born with Significant Cutaneous Hypopigmentation

*Dct^−/−^* pups were easily recognizable from their dark *Dct^+/−^* littermates at P4, as their first hair is significantly paler. At this stage, the tail and outer ear border skin was still unpigmented in the mutants (Figure 2a).

Electron microscopy examination of head skin biopsies evidenced melanosomes in dermis and epidermis melanocytes, as well as in keratinocytes in *Dct^−/−^* samples, yet they were found in reduced numbers and mostly immature with incomplete melanization and irregular contours compared to the controls (Figure 2b). At earlier stages, when the skin is still glabrous (birth to P2), littermate pups of each genotype *Dct^+/−^* and *Dct^−/−^* could not be phenotypically distinguished. However, imaging of the first hair growth on skin flat mounts (hypodermis facing the objective) revealed a significant lack of pigment in the hair follicles, although the number of pigmented hair follicles was not statistically different (Figure 2c and Appendix A). On the other hand, earlier at birth (P0.5), around 50% of hair follicles had not started to become visibly pigmented in *Dct^−/−^* samples, and the overall pigmentation (hair follicles and hypodermis) of the flat mounts was much lighter (Figure 2d).

All in all, these observations showed that the cutaneous hypopigmentation of the *Dct^−/−^* mouse is much more pronounced in the early stages of life than it is in the adult mouse [12,23], probably due to fewer and/or immature melanosomes production in skin melanocytes. However, the delay in the onset of hair follicle melanization is mostly overcome in a few days resulting in a homogeneous dark gray *slaty*-like coat color.

### 3.2. RPE of Dct^−/−^ Mice Is Severely Hypopigmented from Early Embryonic Stages

We next investigated pigmentation of the eye, focusing more specifically on the neuroectoderm-derived RPE. We have previously shown that the RPE of adult *Dct^−/−^* mice is markedly hypopigmented, whereas both the choroid and the iris are as dark as in the C57BL6/J adult controls [12]. *Dct* expression within the developing RPE precedes the expression of *Tyr* and *Tyrp1*. It is detected as early as embryonic day E9.5, i.e., 2 days before the first pigment granules are seen [28]. Here we examined the developing RPE starting with E15.5 embryos when the wild-type non-agouti RPE is homogeneously pigmented. This revealed marked hypopigmentation of *Dct^−/−^* embryonic eyes with some RPE cells still devoid of visible pigment (Figure 3a).

In pigmented cells, pigment granules were distributed as evenly as in controls across the apico-basal axis, different from the apical accumulation reported for embryonic *Tyr^c/c^* RPE melanosomes [8]. Hypopigmentation of the epithelial layer extended anteriorly to the ciliary margin zone (CMZ) at a time when the RPE in this region is known to regulate critical steps of retinogenesis such as retinal mitosis and the specification of ventrotemporal retinal ganglion cells [7,29,30]. Still, all RPE cells in the CMZ of *Dct^−/−^* embryos had visible pigment, and, as in controls, pigment granules were mostly located at the apical pole in the RPE tip cells.

In wild-type, the choroid is not visibly pigmented until after P3 so that the melanized RPE can be seen through the sclera of whole eyeballs collected from newborns. At birth (P0.5), the lack of pigment in the *Dct^−/−^* RPE was obvious (Figure 3b) and could be used as a fully reliable indicator of genotype. Examining flat-mounted eyecups by bright-field microscopy indicated a clear reduction in pigmented melanosomes in *Dct^−/−^* P0.5 RPE compared to littermate control RPE. In the latter, heavily pigmented melanosomes were seen densely packed and pushed against the cell membranes, thus revealing, without requiring staining, the typical honeycomb organization of the cell layer. In order to gain more insight into RPE defects in *Dct^−/−^* mice, ultrastructural analysis was conducted at 1 month when melanosome maturation can be assessed in both RPE and choroid (Figure 3c,d). Parameters that had been described as affected in pigment-deficient models were examined. We were unable to detect any anomalies at the interface between the RPE and neural retina. Phagocytosed photoreceptor segments were detected in similar quantities in *Dct^−/−^* and *Dct^+/−^* RPE. The subcellular distribution of the melanosomes along the apico-basal axis was not visibly disrupted in *Dct^−/−^* RPE (Figure 3c). The proportions of round and elliptic melanosome shapes were also equivalent, illustrating the random orientation of the oblong pigmented eumelamosomes as in controls. By contrast, the total number of pigmented (stage III and IV) melanosomes was significantly reduced in *Dct^−/−^* RPE reaching only 58% of the controls (Figure 3d). Moreover, around 30% of pigmented melanosomes had not reached stage IV (full pigmentation), whereas 98% were at stage IV in the controls. Among stage IV melanosomes scored in *Dct^−/−^* RPE, over 40% harbored irregular contours, contrasting with the smooth appearance of the control stage IV melanosomes. The density and ultrastructure of melanosomes in the neighboring choroid of these samples were also analyzed. A few fields showed elevated amounts of immature stage III melanosomes, and the melanosome density seemed to be reduced in these same fields (Appendix A). In most of the choroidal tissue, however, melanosome counts and staging were not significantly different from controls, correlating with the overall dark choroid appearance already observed in adult *Dct^−/−^* mice [12,23].

All in all, while *Dct* loss of function has only a moderate long-lasting effect on melanocyte pigmentation, whether, in the skin or choroid, its impact on initial melanization of the RPE is striking and is not overcome after birth [12]. In contrast to the *albino* models, the embryonic Dct-deficient RPE is slightly pigmented at E15.5, including in the CMZ, where major regulation of neuron cell fate occurs and is believed to depend upon L-Dopa.

### 3.3. The RPE Cells of Dct^−/−^ Neonates Show Increased Size and Junction Defects

Cell shape is modified, and average cell size is increased in the RPE of postnatal *albino* rodents [31]. We, therefore, asked if the general cell organization and morphology were disrupted in the RPE of *Dct^−/−^* newborns, which could indicate developmental anomalies similar to what has been described for the embryonic RPE of the *Tyr^c/c^* line [8]. To this end, flat mounts of RPE prepared from P0.5 mice were stained with phalloidin that binds F-actin specifically located at the apical borders in RPE cells [32] (Figure 4 and Appendix A).

We conducted the sampling and analysis at P0.5 on RPEs from *Dct^+/−^* littermates as normally pigmented controls as well as *Tyr^c/c^* RPEs of age-matched coisogenic mice. Average cell size was visibly increased in P0.5 *Tyr^c/c^* RPE in all examined samples, as illustrated by a representative image (Figure 4b) and as expected [8]. At the same age, *Dct^−/−^* RPE cells were also larger than controls (Figure 4c). Quantification of the average cell surface for one representative RPE of each genotype indicated that this increase in cell size was significant for both *Tyr^c/c^* and *Dct^−/−^*, yet significantly less pronounced in *Dct^−/−^* newborns than in *Tyr^c/c^* (Figure 4d). Cell size and shape were visibly more heterogenous in *Dct^−/−^* RPE than in controls, as illustrated by representative fields in Figure 4. Thus, the irregular RPE cell morphology displayed by *Dct^−/−^* newborns was similar to what has been described in *Tyr^c/c^* late embryos [8].

In the same series of observations, Iwai-Takekoshi et al. showed that both gap and adherens RPE cell junctions are profoundly disrupted in the absence of Tyr. In particular, the labeling of P-cadherin, the major cadherin of mammalian RPE [33], evidenced a loose distribution of the adherens junctional protein in *Tyr^c/c^* embryos [8]. We, therefore, analyzed P0.5 flat mounts of *Dct^−/−^* RPEs in parallel to *Dct^+/−^*and *Tyr^c/c^* RPEs after staining with anti-P-cadherin antibodies (Figure 5).

As illustrated, P-cadherin labeling revealed an even outline of the cell membranes of control P0.5 *Dct^+/−^* RPE cells, with well-defined sides and sharp corners of each polygon (Figure 5a). By contrast, in *Tyr^c/c^* P0.5 RPE cells (Figure 5b), and as expected from the embryonic pattern [8], P-cadherin was found diffusely distributed, resulting in a tortuous appearance of the cell borders. Although less pronounced, a similar anomaly was observed in *Dct^−/−^* RPE flat mounts (Figure 5c). In these, P-cadherin immunodetection highlighted irregular cell shapes, tortuous sides, and poorly defined corners of the polygons, as well as diffuse staining under the apical membrane suggesting a junctional defect comparable to what has been concluded for the *Tyr^c/c^* model. Interestingly, the larger cells in *Dct^−/−^* and *Tyr^c/c^* RPE are equally multinucleated. Similar observations have been made in the *albino* rat, in which a population of RPE cells is retained in the cell cycle without achieving cell division [31]

Thus, the characterization of neonatal *Dct^−/−^* RPE indicates that, although mature stage III/IV melanosomes are present and correctly distributed along the apico-basal axis, loss of function of *Dct* results in disruption of cell junctions and morphology as described for *Tyr* is mutants. This suggests that essential intracrine developmental regulators are impaired in the absence of Dct as when tyrosinase activity is lacking. Hydroxylation of the initial substrate tyrosine by tyrosinase results in L-Dopa production (Figure 1) that is converted to dopachrome, the substrate of Dct, in eumelanosomes. L-Dopa produced in the developing RPE is believed to influence one or several critical processes of retinogenesis that are impaired in patients and models of albinism, including RPE cell division [10,31,34,35]. As Dct acts downstream of Tyr, the loss of enzymatic activity is not expected to disrupt L-Dopa production unless the excess of dopachrome exerts a negative retrocontrol on Tyr or if Dct is required for sufficient tyrosinase activity in the tightly regulated developmental context.

### 3.4. L-Dopa Concentration Is Reduced in Postnatal Eyecups of Dct^−/−^ Mice

The kinetics of L-Dopa production in both fetal and postnatal C57BL/6J mouse eyecups has been described [11]. It follows three phases: 1—a slow rise in L-Dopa production from E10.5 until P2 when tyrosinase is active in the RPE only; 2—a ~10-day period of more pronounced increase correlating with melanogenesis induction in the choroidal melanocytes; 3—a gradual drop to initial levels.

We compared the concentration of L-Dopa in postnatal eyecups of *Dct^+/−^* and *Dct^−/−^* postnatal eyes between P6 and P20, i.e., during the period of choroidal melanization (Figure 6).

As expected, L-Dopa was not detected in samples from *Tyr^c/c^* control mice (not shown). By contrast, L-Dopa was detected in substantial amounts in *Dct^−/−^* eyecups, and the concentration gradually increased over the studied period with a comparable slope as in *Dct^+/−^* control samples. However, at each time point, the level of L-Dopa in *Dct^−/−^* samples was significantly lower than in control littermates. For each genotype, the slopes of the calculated linear regressions were significantly different from zero (*p* < 0.05) and not significantly different from each other, with *Dct^−/−^* eyecups producing only 54% of the amounts detected in *Dct^+/−^* controls at P6. So, although the choroids of *Dct^−/−^* mice become pigmented over time and can not be distinguished from the choroid of controls by the apparent pigmentation of the eyeball, we show that the content of L-Dopa is significantly reduced in postnatal mutant eyecups, whether in the choroid, RPE or both. These results suggest that, unexpectedly, L-Dopa may be limiting due to Dct inactivation in the context of retinal development.

## 4. Discussion

### 4.1. The Identification of OCA8 Patients Gives Rise to a Renewed Interest for Dct^−/−^ Animal Models

The first *Dct* knockout mouse was published in 2004 at a time when no human *DCT* pathogenic variants had yet been identified [23]. This model was found to be fully viable and fertile with no additional anomalies than their lighter coat, ruling out *Dct* as essential, in mice at least, despite its transient expression in the embryonic telencephalon [28]. The model was exploited further in order to test the implication of Dct in skin protection against UVA radiation [36]. The study confirmed the role of melanocytic Dct in the regulation of DHICA-mediated antioxidation in vivo. OCA patients with biallelic pathogenic variants of *DCT* were finally reported only last year [12]. Their severe visual deficit contrasts with moderate skin/hair hypopigmentation. This gives rise to renewed interest in the role of dopachrome tautomerase in pigment cell biology as well as in the exact molecular link between melanogenesis and retinogenesis. Two of the three human pathogenic variants that we reported result in substitutions of cysteine residues (Cys40Ser and Cys61Trp) in the highly conserved N-terminal Cys-rich domain of DCT [12]. Recently, the corresponding recombinant proteins have been produced and characterized, which confirmed that they are both highly unstable, as expected from the loss of the corresponding disulfide bridges [37]. The third variant that we described is an out-of-frame deletion. The corresponding truncated protein lacks the transmembrane domain, which is necessary for anchoring the protein in the melanosomal membrane and is therefore likely a null as well. An additional OCA8 patient identified by Volk et al. is compound heterozygous for two nonsense variants, again with a predicted lack of the transmembrane domain [14]. All in all, human pathogenic variants of *DCT* described so far as associated with OCA8 probably behave as null alleles: the putative encoded proteins are predicted to be highly unstable and/or unable to reach the melanosome, making it unlikely that they would interfere with the melanogenesis machinery. In addition, Volk et al. reported a consanguineous family with three patients homozygous for the *DCT* missense variant c.176G > T (p.Gly59Val) with a possibly less detrimental effect on the function of the encoded protein. Interestingly, only one of the three patients suffered from foveal hypoplasia and mild iris transillumination in addition to infantile nystagmus, while his two cousins (brother and sister) were diagnosed with nystagmus and reduced tanning only [14]. These observations argue for a dose-dependent effect of DCT, which might therefore prove limiting in the context of finely tuned retinogenesis. In order to obtain more insight into the molecular mechanisms that link DCT-mediated melanogenesis and retinal development, we need a relevant Dct-deficient animal model in which the RPE can be experimentally investigated.

Earlier interpretations of the role of Dct in melanogenesis were mainly based on *Slaty* (*Dct^slt^*/*Dct^slt^*) mice and/or their melanocytes derived in vitro [38,39]. There is concern, however, that the *Dct^slt^* allele is not a null mutation, which may hinder functional interpretation. For instance, the hypopigmentation of *Slaty* mice has been described to worsen with age and is accompanied by a significant premature loss of the first hair on different genetic backgrounds, including non-agouti C57BL [40]. Neither Guyonneau et al. nor we observed such a phenomenon in our genetically engineered loss-of-function models. Moreover, in some melanin dosage assays, the two genotypes show significant differences, with pigmentation being unexpectedly more affected in *Slaty* hair than in the knockout hair [23]. At the level of melanosome structure, *Slaty* and *Slaty^Light^* melanocytes produce immature and/or abnormal melanosomes, with *Slaty^Light^* lacking homogeneously pigmented stage IV melanosomes [39]. This melanosomal phenotype is worse than what we observed in the skin/choroidal melanocytes of our *Dct^−/−^* mice (Figure 2 and Appendix A) or in MNT-1 cells with a *DCT* knockout (V.M., E.M., A.T., S.J., unpublished results). In both models, we detect a majority of stage IV melanosomes, suggesting that *Slaty* and *Slaty^Light^* mutant Dct proteins interfere with melanosomal maturation and/or integrity, masking the direct effect of their low enzymatic activity. In order to link melanogenesis and retinogenesis, we need to learn if L-Dopa production depends on Dct, for instance, via Dct/Tyr interactions. When *Slaty* or *Slaty^Light^* melanocytes are grown in culture, although the activity of Dct is found to be much reduced (36% and 4%, respectively), the level of activity of tyrosinase is not impacted [39]. Still, the mutant Dct (*Slaty* or *Slaty^Light^*), which is correctly addressed to the melanosome, could contribute to tyrosinase trafficking and/or stability, independent of its own enzymatic activity defect, which leaves the question of wild-type Dct/Tyr interactions unanswered.

As mentioned above, OCA8 patients with a loss of function of *DCT* suffer from visual deficits equivalent to other types of OCA. The key role of the RPE has been extensively highlighted in OCA1 *albino* models, while this tissue has not even begun to be explored neither in *Slaty* or *Slaty^Light^* mice nor in knockout mice. For this reason, we mainly focused our interest on the RPE of *Dct^−/−^* mice and asked if we could identify phenotypes similar to what has been described in *Tyr^c/c^* mice, with special emphasis on melanosomes, cell shape/size, and junction defects.

### 4.2. What Do We Learn from the Dct^−/−^ Mouse and What More Do We Expect to Discover?

In our initial description of the *Dct^−/−^* model, we reported that the absence of Dct has a more visible impact on the pigmentation of the RPE than on that of the melanocytic tissues (skin/choroid/iris) [12]. Here we show that, at the onset of melanosomal biogenesis, there is a comparable lack of pigment in the skin and RPE of *Dct^−/−^* mice, with a reduced amount of mature melanosomes in both cell types. In wild-type mouse melanocytes, melanosome biogenesis is sustained after birth, including in the choroid, whereas in the RPE, melanosome biogenesis is achieved predominantly in a short embryonic time window. Moreover, RPE cells are essentially post-mitotic, for review [41]. This could account for the more persistent melanosomal phenotype in the RPE cells of *Dct^−/−^* mice compared to that of their melanocytes. Interestingly, a similar interpretation has been proposed to explain the marked RPE phenotype of *Rab38^−/−^*, also known as *chocolate* mice, that are deficient in melanosome maturation [42]. We showed that at E15.5, i.e., when the RPE is believed to exert an essential role on retinal ganglion cell specification [7,29,30], the lack of RPE pigment was already evident on the whole surface of the RPE, including the ciliary margin. The striking hypopigmentation of the *Dct^−/−^* RPE was best evidenced at birth on whole enucleated eyeballs when the choroid that covers the RPE layer is not yet pigmented. At that stage, the marked contrast between the RPE, which was nearly devoid of pigment, and the black iris rim, was very similar to what has been described in newborns of the *dark-eyed albino* line. This line is homozygous for a hypomorphic allele of *Tyr* (*Tyr^c44H^*), resulting in a reduction in tyrosinase activity to around 2% of normal [43]. Moreover, the marked pigment deficit of *Dct^−/−^* RPE was associated with anomalies in cell shape and cellular junctions comparable to those detected in *Tyr^c/c^* tissues. In particular, the cells appeared more heterogeneous in size and generally larger, and P-cadherin, which labels the adherens junctions, was loosely distributed. Although we have not yet explored these defects as precisely as reported for *Tyr^c/c^* [8], we can conclude that the RPEs of *Dct^−/−^* and *Tyr^c/c^* mutants share some major defects in cell organization. What molecular regulators could be affected in this same way by both genotypes? L-Dopa, which is deficient in *Tyr^c/c^* mutants, is suspected to be instrumental in retinogenesis [10,44]. Although the paradigm of melanin biosynthesis argues against Dct being involved in L-Dopa production, the protein may determinately contribute, at least in specific contexts such as finely tuned development, to trafficking and/or activation and/or stabilization of tyrosinase and therefore to the release of sufficient amounts of L-Dopa to drive retinal maturation. Supporting the hypothesis that the three melanogenic enzymes interact directly within the melanosome, recent 3D structural models based on the crystallographic structure of human TYRP1 suggest the possible assembling of TYR, TYRP1, and DCT into heterodimeric (TYR-DCT; TYR-TYRP1) or heterotrimeric complexes [17]. Here we show that the level of L-Dopa is significantly reduced in postnatal eyecups of *Dct^−/−^* mice during melanization of the choroid. Although negative feedback driven by accumulated dopachrome is not excluded, our observation adds credit to the hypothesis that Dct contributes, at least in certain contexts, to L-Dopa production in vivo. We suggest that Dct can activate or stabilize Tyr, including in the developing RPE. Investigating earlier stages from midgestation to P2 before melanogenesis is induced in the choroid will tell us if L-Dopa is indeed in too small a quantity in the Dct-deficient RPE at a timing that is critical for retinal neurogenesis. Thus and quite unexpectedly, L-Dopa deficiency in the embryonic RPE may prove to be a common denominator to OCA1 and OCA8, raising the question of the same misregulation occurring in other types of albinism such as OCA3.

This study paves the way for further comparative characterization of the developing RPE and neural retina of the *Dct^−/−^* and *Tyr^c/c^* mice. Particular attention should be paid to the genesis of RPE-specific defects such as cell shape and multinucleated cell distribution that we have observed in postnatal *Dct^−/−^* RPE. The identification of common and specific molecular/cellular phenotypes will guide future investigations, not only during development when the retinal defects are first induced but also in the adult, in the course of aging. The iris transillumination and pigment dispersion in *Slaty^Light^* have been shown to worsen with age [45]. This could be due to the disruption of melanosomes induced by the *Slaty^Light^* encoded protein that may cause the release of toxic melanin intermediates such as dopachrome. Whether a similar degenerative profile is observed in *Dct^−/−^* mice remains to be determined. These future investigations in the development and aging of pigmented tissues will have important implications for clinical follow-up and prevention dedicated to OCA8 patients taking into account their *DCT* genotype.

## 5. Conclusions

Here we show that the *Dct^−/−^* mouse RPE displays severe developmental defects similar to what has been described in the *Tyr^−/−^ albino* mouse. These are likely associated to a lack of L-Dopa at critical stages of retinogenesis. Future work will aim at confirming L-Dopa deficit in the embryonic *Dct^−/−^* RPE and evaluating the consequences on the neural retina, especially on photoreceptor distribution and chiasmal misrouting. These findings should have important implications for the prognosis of retinal pathology in patients with albinism according to their genetic diagnosis.

## Figures and Tables

**Figure 1 genes-13-01164-f001:**
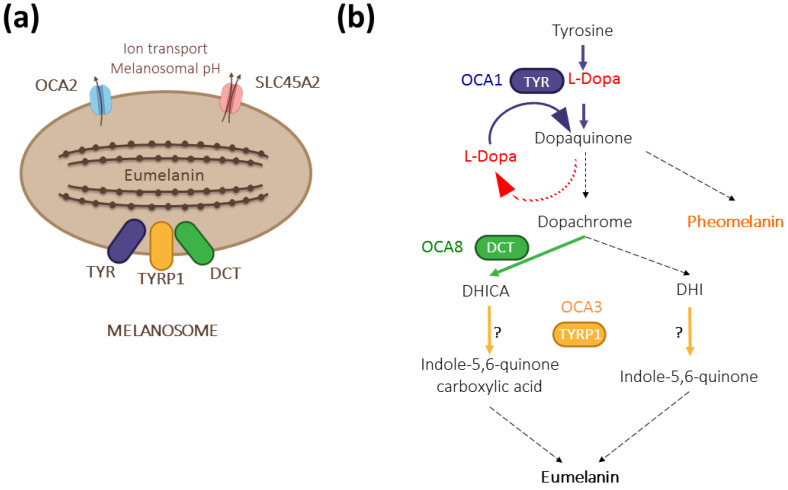
The contribution of DCT/TYRP2 (dopachrome tautomerase/tyrosinase-related protein 2) to melanogenesis. (**a**) Together with its paralogs TYR (tyrosinase) and TYRP1 (tyrosinase-related protein 1), DCT is anchored at the membrane of eumelanin-producing melanosomes with its enzymatic domain localized inside. The formation of a multienzymatic complex has been suggested [16,17] but has not been formally proven in vivo. Other major albinism genes, such as *OCA2* (OCA2) and *SLC45A2* (OCA4), encode regulators of melanogenesis by controlling the pH within the melanosome [18]. (**b**) As a melanogenic enzyme, DCT acts downstream of TYR and contributes to dark pigment synthesis by converting dopachrome to DHICA. TYR is the rate-limiting enzyme with both its hydroxylase and oxidase activities regulating the amount of L-Dopa in pigmented cells. TYRP1 may oxidize DHICA and/or DHI depending on the context [19,20]. Dashed arrows indicate spontaneous reactions.

**Figure 2 genes-13-01164-f002:**
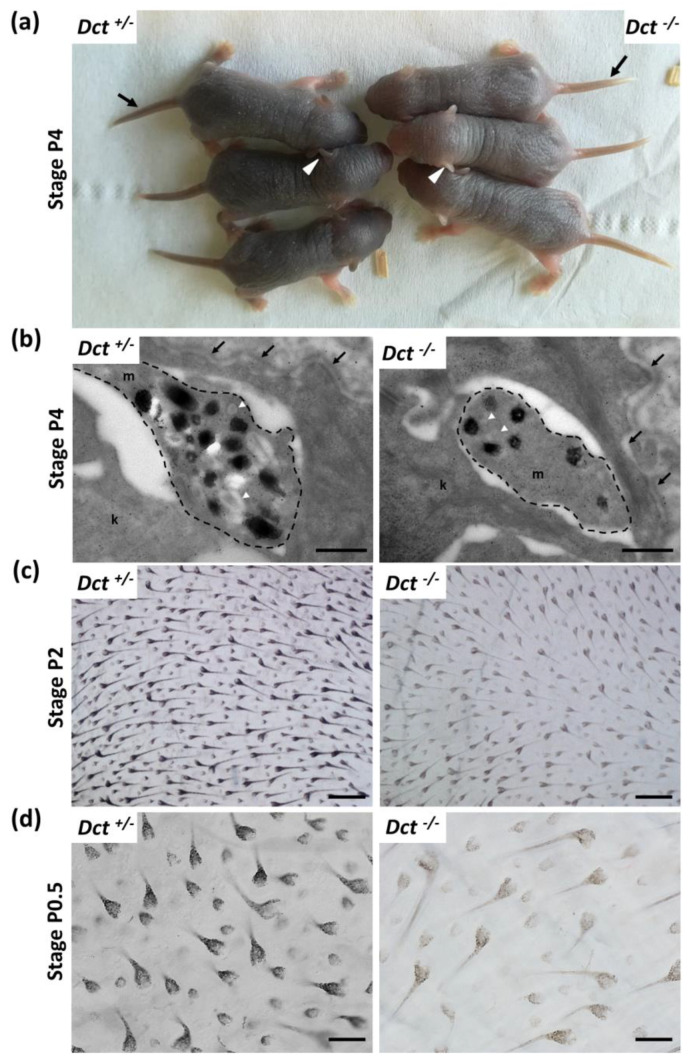
Melanization of the skin and hair in newborn *Dct^−/−^* mice. (**a**) *Dct^−/−^* pups can be easily distinguished from *Dct^+/−^* littermates at P4 when their skin and first hair are visibly lighter. The tail (black arrow) and the outer ear (white arrowheads) are unpigmented in *Dct^−/−^* pups at this stage, whereas the tail and ears of control littermates are dark; (**b**) ultrastructure of epidermal melanocytic extensions (marked out with a dashed line) in the skin of P4 mice, showing that *Dct^−/−^* skin has fewer stage III/IV melanosomes than controls. Most pigmented melanosomes have irregular borders in *Dct^−/−^* epidermis. k, keratinocyte; m, melanocyte; the dermis-epidermis junction is indicated by arrows. Early unpigmented melanosomes are indicated by white arrowheads; scale bar: 500 nm; (**c**) bright-field microscopy imaging of flat-mounted skin biopsies at P2 showing pigmentation of the first hair follicles at the inner face (for quantification, see Appendix A); scale bar: 250 μm; (**d**) same imaging as at P0.5 when only 50% of hair follicles are visibly pigmented in *Dct^−/−^* pups; note that the hypodermis is lighter in *Dct^−/−^* samples; scale bar: 100 μm.

**Figure 3 genes-13-01164-f003:**
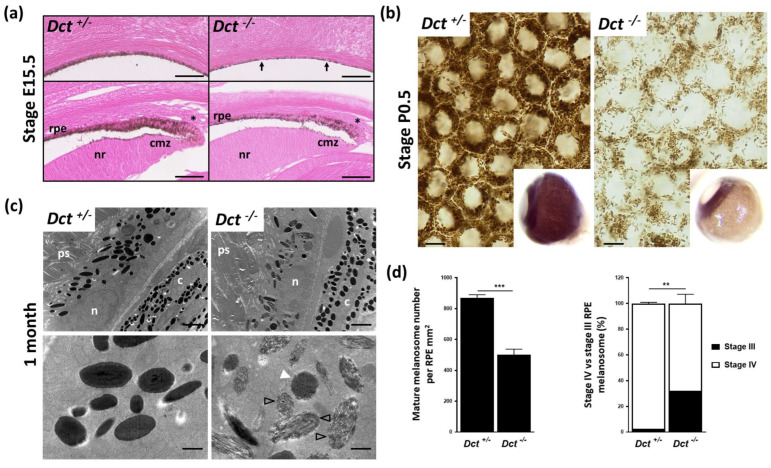
Defect in melanogenesis in the developing RPE of *Dct^−/−^* mice. (**a**) Eosin histology of the E15.5 eye showing the RPE in equatorial (upper panels) and anterior (lower panels) fields; scale bar: 50 μm. Arrows point to unpigmented cells in the RPE of a *Dct^−/−^* embryo, whereas at the same stage, all RPE cells are pigmented in the control embryo; Asterisks indicate tip RPE cells with pigment granules located apically in both *Dct*^−/−^ and control eyes; cmz, ciliary margin zone; nr, neural retina; (**b**) Bright-field microscopy of P0.5 RPE flat mounts showing a reduced number of pigment granules in *Dct^−/−^* samples, scale bar: 10 μm. Inserts show whole eyeballs. At this stage, only the iris and RPE are pigmented in the wild-type, not the outer layer (choroid), so that hypopigmentation of the RPE can be readily detected on enucleated *Dct^−/−^* eyes; (**c**) Transmission electron microscopy of the RPE at 1 month. Low magnification (upper panels, scale bar: 2 μm) shows no difference in the subcellular localization and the shape (round vs. elliptic) distribution of the melanosomes, but these are fewer and less electron-dense in *Dct*^−/−^ RPE; c, choroid; *n*, RPE cell nucleus; ps, photoreceptor segment. High magnification (lower panels, scale bar: 500 nm) shows that most of the melanosomes in *Dct*^−/−^ RPE are at stage III (black arrowheads) or stage IV but with irregular contours (white arrowhead) compared to the controls that are mostly stage IV and exhibit a smooth shape; (**d**) Quantification of mature melanosomes (left histogram) and stage IV vs. stage III melanosomes (right histogram) from representative sections of one eye per genotype. Total number of analyzed melanosomes: *n* > 2000 for *Dct*^+/−^, *n* > 1000 for *Dct*^−/−^. Values are presented as means ± SEM. Means were compared using *t*-test, ** *p* < 0.01; *** *p* < 0.001.

**Figure 4 genes-13-01164-f004:**
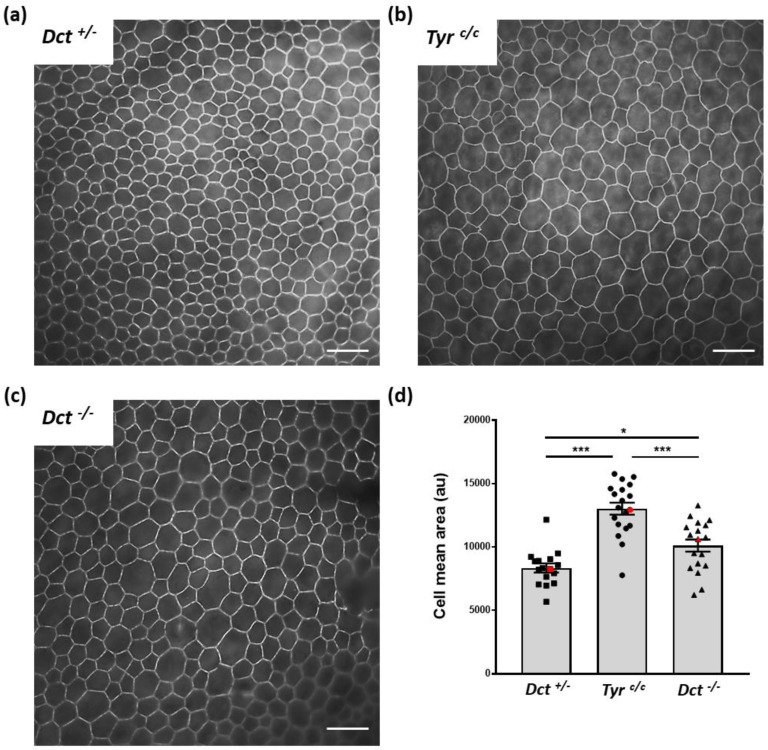
RPE cell size and shape at P0.5. Total RPE flat mounts (for each genotype *n* = 6; 2 RPEs from 3 mice) were labeled with Alexafluor 488 phalloidin (see whole samples at low magnification, Appendix A), and contiguous images of the whole surface were acquired through a ×40 oil immersion lens for one representative eye per genotype, scale bar: 25 μm. A representative image for each genotype is shown. (**a**) *Dct^+/−^* used as wild-type littermate control; (**b**) *Tyr^c/c^* used as age-matched non-littermate albino control displaying bigger cells than *Dct^+/−^*; (**c**) *Dct^−/−^* also displaying bigger cells than *Dct^+/−^*; (**d**) the cell size distribution and average were estimated by manually scoring all cells in each contiguous images from both left and right RPEs of one animal per genotype (number of fields: *Dct^+/−^ n*= 16, *Tyr^c/c^ n*= 19, *Dct^−/−^ n*= 18). The deduced average cell size was plotted for each field. Red symbols correspond to the mean values calculated from selected fields in 4a, 4b, and 4c. The mean ± SEM is indicated by histograms. Significance of differences between groups was determined by one-way ANOVA followed by the Bonferroni post-hoc multiple comparison test, * *p* < 0.05; *** *p* < 0.001.

**Figure 5 genes-13-01164-f005:**

P-cadherin labeling of newborn (P0.5) RPE. RPE flat mounts were immunostained with anti-P-cadherin antibodies that label the adherens junctions in green, scale bar: 10 μm. Nuclei were stained with DAPI and are shown in red. Each picture is a stack of 4 to 6 Z-planes (0.2 μm) for flatness compensation and volume rendering; (**a**) *Dct^+/−^* used as wild-type littermate control; (**b**) *Tyr^c/c^* used as age-matched non-littermate albino control; (**c**) *Dct^−/−^*. This labeling reveals abnormal adherens junction in *Dct^−/−^* mice similar to *Tyr^c/c^.* Note that the bigger cells in *Tyr^c/c^* and *Dct^−/−^* are mostly multinucleated (nuclei oulined with white dotted lines).

**Figure 6 genes-13-01164-f006:**
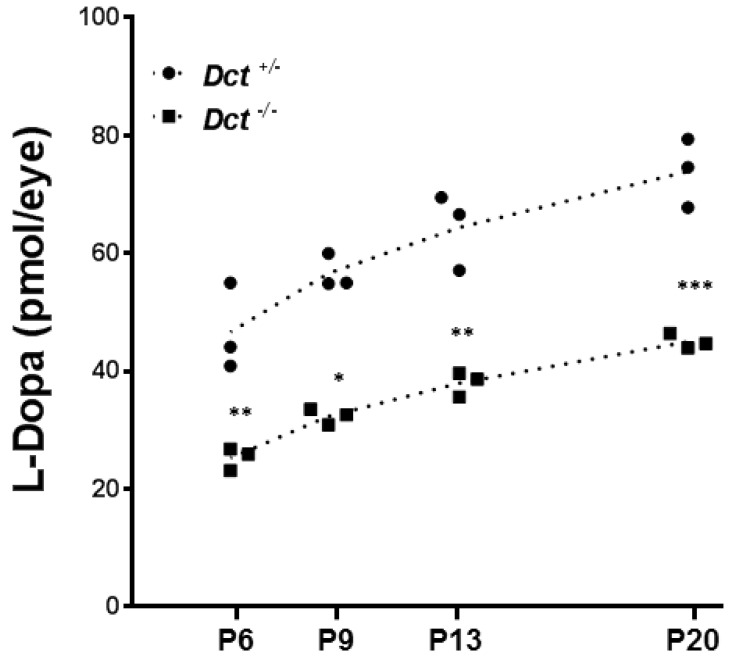
Dosage of L-Dopa in postnatal eyecups. L-Dopa was quantified by HPIC in *Dct^−/−^* eyecups (square symbols)) compared to *Dct^+/−^* (black dots). Eyecups from *Tyr^c/c^* mice were used to define the baseline. At each indicated postnatal day: P6, P9, P13, P20, both eyes from 3 littermates of each genotype were dissected, pooled, and used for dosage. For each stage of development, L-Dopa quantification for both genotypes were compared by using unpaired *t*-test,* *p* < 0.05, ** *p* < 0.01; *** *p* < 0.001.

## Data Availability

Not applicable.

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
