# Peer review of "The Dct−/− Mouse Model to Unravel Retinogenesis Misregulation in Patients with Albinism"

_genes, 2022, doi:10.3390/genes13071164_

Round 1
Reviewer 1 Report
This is a sound and interesting study following up the previous description of Dct mutant mice, generated by the same authors with mutations in the human DCT gene previously found and associated with OCA8 albinism type. The authors nicely delineate the subcellular and cellular alterations of Dct mutant mice in skin melanocytes and RPE cells during late development. Their analyses clearly show cell morphology alterations, number of melanosomes and density distorted and decreased levels of the metabolite intermediate L-DOPA in this animal model, as compared to pigmented controls. Although not yet explained, the results obtained in this work suggest an interaction between melanogenic enzymes that might account for the observed L-DOPA decreased accumulation in Dct mutant mice, taking into account Dct enzyme acts downstream of Tyr, where L-DOPA production is associated. In all respects, this is an interesting scientific study shedding light to the origin of visual alterations associated with albinism commonly found in all albinism types, including the most recently described OCA8 type.
Author Response
We thank this reviewer for his positive comments on the manuscript.
Reviewer 2 Report
This manuscript by Tingaud-Sequeira and colleagues is a well-written and interesting phenotypic characterization of oculocutaneous albinism type 8 in the mouse. This mutation was recently characterized in humans and modeled in mouse by the senior authors Benoit Arveiler and Ian Jackson, and Sophie Javerzat, the latter senior author of the present study (ref 12).
The pigment loss phenotype of OCA8, cue to mutation in dopachrome tautomerase, is obvious in skin and eye early in embryogenesis, then is partially restored in skin, but less so in the RPE, which remains hypopigmented. The mouse and its phenotype are therefore good models for understanding melanogenesis and the role of melanin in eye development using this new mouse model and form of albinism.
Major comments:
1. The title of the paper prepares the reader to read about retinogenesis (i.e., retinal development, or perhaps, neurogenesis), but there is little data on these aspects. The paper instead describes defects in the RPE in OCA8. The title should thus be changed.
2. There is scant allusion in the Introduction on why this study is significant. How will these data and very fine descriptions contribute to the field? At the very end, in the last paragraph of the Discussion, there is mention that the study “paves the way” for further comparison of RPE/neural retina in Dct-/- and Tyrc/c mice and that more cellular/molecular features will be interrogated in development and aging. These statements could be made stronger and earlier.
3. Have the authors considered performing functional/behavioral tests to prove the relevance and even utility of this mutation, to study albinism? For example, are there plans to test photosensitivity, pupillary reflex, visual acuity/contrast sensitivity in these mice?
4. Results:
a. Figure 2 a, c, d: the contrast and brightness are different comparing het and mutant; and in a. the illumination is uneven and thus, comparisons are difficult to make.
b. Figure 3 a: Likewise, the panels for het and wt are of different brightness and should be made even. Also the white space dividing upper and lower is difficult to see; add a black line?
c. P.9, final paragraph of section 3.2: The time line of less to more melanization is confusing. The sentence “All in all, ….” Should be reqritten
i. A table indicating melanin/melanocyte content in skin vs eye, over the several times examined would be helpful.
ii. The age should be indicated on the micrographs directly for clarity.
d. Figure 4:
i. The statistical significance/comparisons should be indicated on d.
ii. The difference in size in the micrographs is difficult to see.
iii. Fields of each eye needs to be averaged; the fields of each of a single animal should be averaged, so that n = animals. Otherwise this is overrepresentation of the data.
e. Multinucleation:
i. The micrographs should be presented at higher magnification and higher resolution.
ii. P. 11, second paragraph: The differences should be played down as the RPE even in the pigmented mouse can be multinucleated.
f. L-Dopa: It is fascinating that L-Dopa levels are reasonably high in the Dct-/- retina. In the paragraph just above Section 3.5, there is mention of dopachrome exerting a “retrocontrol” on Tyr/tyrosinase activity. This should be highlighted in the Discussion as well. Also “negative feedback” is perhaps a better term than retrocontrol.
5. The authors might want to cite a recent paper by George…Bharti and Brooks, Stem Cell Reports, 2022, “In vitro disease modeling of oculocutaneous albinism type 1 and 2 using human induced pluripotent stem cell-derived retinal pigment epithelium”, for comparison with their melanocyte phenotypes.
6. What does Figure S3 show? Phalloidin staining nicely shows the outlines of the RPE cells but the retinal wholemount staining might be highlighting axons (?).
Minor edits:
1. P. 3, last line, “evidence” should be changed to “demonstrate” or some other verb.
2. Just before Discussion, “retinogenesis” should be changed to “retinal development”.
Author Response
Dear reviewer,
We have now carefully addressed your comments and revised the manuscript accordingly. Please find attached our point-by-point response detailing the changes that you will find in the modified submitted version.
